# Post-translational toxin modification by lactate controls *Staphylococcus aureus* virulence

Yanan Wang [1,2,5], Yanfeng Liu [3,5], Guoxiu Xiang[1,5], Ying Jian[1], Ziyu Yang[1], Tianchi Chen[1], Xiaowei Ma[1], Na Zhao[1], Yingxin Dai[1], Yan Lv[1], Hua Wang[1], Lei He [1], Bisheng Shi[1], Qian Liu[1,4], Yao Liu[1,4], Michael Otto [4] ✉ & Min Li [1,2] ✉

Diverse post-translational modifications have been shown to play important roles in regulating protein function in eukaryotes. By contrast, the roles of post-translational modifications in bacteria are not so well understood, particularly as they relate to pathogenesis. Here, we demonstrate post-translational protein modification by covalent addition of lactate to lysine residues (lactylation) in the human pathogen *Staphylococcus aureus*. Lactylation is dependent on lactate concentration and specifically affects alpha-toxin, in which a single lactylated lysine is required for full activity and virulence in infection models. Given that lactate levels typically increase during infection, our results suggest that the pathogen can use protein lactylation as a mechanism to increase toxin-mediated virulence during infection.

Bacterial infections represent one of the leading global causes of death[1]. *Staphylococcus aureus* is one of the most frequent causes of mortality by an infectious bacterial agent[1]. As in many other bacterial pathogens, infectivity of *S. aureus* is largely based on toxin production[2]. *S. aureus* secretes a vast array of toxins, among which alpha-toxin is arguably the most important in terms of contribution to pathogenesis and the best studied[3,4]. It is also the toxin most frequently investigated for *S. aureus* anti-virulence strategies, such as by using anti-toxin monoclonal antibodies[5–7]. Alpha-toxin exerts its pathogenic role predominantly via the cytolysis of a variety of cell types, including erythrocytes, leukocytes, endo- and epithelial cells, in a mostly receptor-dependent fashion[4]. This distinguishes alpha-toxin from the other main toxin families in *S. aureus*, the bicomponent leucocidins, superantigenic toxins, and phenol-soluble modulins (PSMs), which are either target cell-specific or work in a receptor-independent fashion[3,8–10].

Post-translational modification (PTM) of proteins is an important means to alter the function of a protein[11,12]. Proteolytic cleavage and cystine bridge formation are examples of PTM that are often required to form the mature form of an enzyme or hormone. However, in a stricter sense, this term more often refers to the covalent addition of moieties to the N- or C-termini or the amino acid side chains of proteins. Many different moieties can be linked to proteins in such a fashion, with the most frequent modifications being phosphorylation, acetylation, and glycosylation[13]. PTM has been believed to occur more frequently and play a more pronounced role in eukaryotes than prokaryotes, but there also is increased recent attention to PTM in bacteria[14,15]. However, except for the well-described PTM of ribosomally synthesized bacteriocins[16], which kill other bacteria, it has remained poorly understood whether PTMs have a significant impact on the function of bacterial proteins. As for bacterial protein toxins, functional dependence on PTM has only been reported for the alpha-hemolysin of *Escherichia coli*[17] and the adenylate cyclase toxin of *Bordetella pertussis*[18], which are activated by fatty acylation. Notably, it is not clear whether there is a link between toxin activation by PTM and the environmental conditions during infection.

[1]Department of Laboratory Medicine, Ren Ji Hospital, School of Medicine, Shanghai Jiao Tong University, Shanghai, China. [2]Faculty of Medical Laboratory Science, College of Health Science and Technology, School of Medicine, Shanghai Jiao Tong University, Shanghai, China. [3]Department of Liver Surgery, Ren Ji Hospital, School of Medicine, Shanghai Jiao Tong University, Shanghai, China. [4]Pathogen Molecular Genetics Section, Laboratory of Bacteriology, National Institute of Allergy and Infectious Diseases, National Institutes of Health, Bethesda, MD, USA. [5]These authors contributed equally: Yanan Wang, Yanfeng Liu, Guoxiu Xiang. ✉e-mail: motto@niaid.nih.gov; rjlimin@shsmu.edu.cn

Lactylation of lysine side chains is an only recently discovered form of PTM. It has been first described in eukaryotes, where it also has been studied almost exclusively. Lactylation has specifically been shown to occur in histones and to increase with the increased tissue concentration of lactate that is triggered by hypoxia or pathogen invasion[19,20]. This process involves dedicated lactylase and delactylase enzymes. In bacteria, protein lactylation has only been reported in 2022. Dong et al. showed lactylation of a series of *E. coli* proteins and its effect on bacterial metabolism and identified the involved lysine lactylase and delactylase enzymes[21]. Furthermore, Li et al. recently determined protein lactylation in *Streptococcus mutans* and also revealed an impact on metabolic processes[22]. However, it is not known whether lactylation in bacteria occurs in bacterial toxins and potentially affects their activity, and generally, whether protein lactylation can impact bacterial pathogenesis.

In the present study, we demonstrate non-random lactylation of proteins in *S. aureus*. We specifically show that alpha-toxin becomes lactylated at several residues, one of which is crucial for full activity and significantly impacts virulence in vivo. Dependence on extracellular lactate links this unexpected finding of function-altering PTM of alpha-toxin to the environmental conditions during infection,

identifying a previously unrecognized mechanism by which pathogenic bacteria can adapt toxicity to the host environment.

## Results

### Protein lactylation occurs in *S. aureus* and particularly affects alpha-toxin

Our initial hypothesis in this project was that lactate levels that are knowingly increased in infected tissues intensify *S. aureus* virulence. Such increase of lactate production has been shown for example in keratinocytes infected with *S. aureus*[23] and macrophages infected with *Mycobacterium tuberculosis*[24], and we show here that A549 alveolar epithelial cells infected with live *S. aureus* or exposed to *S. aureus* culture filtrate also produce increased concentrations of lactate (Fig. 1a). For this and further experiments in our study, we used a strain of the ST398 background, a lineage with pronounced clinical importance[25–27], which was selected among several strains based on its average hemolysis capacity to avoid using a strain with uncommon features (Supplementary Fig. 1a, b). To evaluate our hypothesis, we measured cytolytic activity toward sheep red blood cells (RBCs) exerted by *S. aureus* with increasing concentrations of lactate in the media. Cytolysis of RBCs (hemolysis) is a key virulence mechanism of *S.*

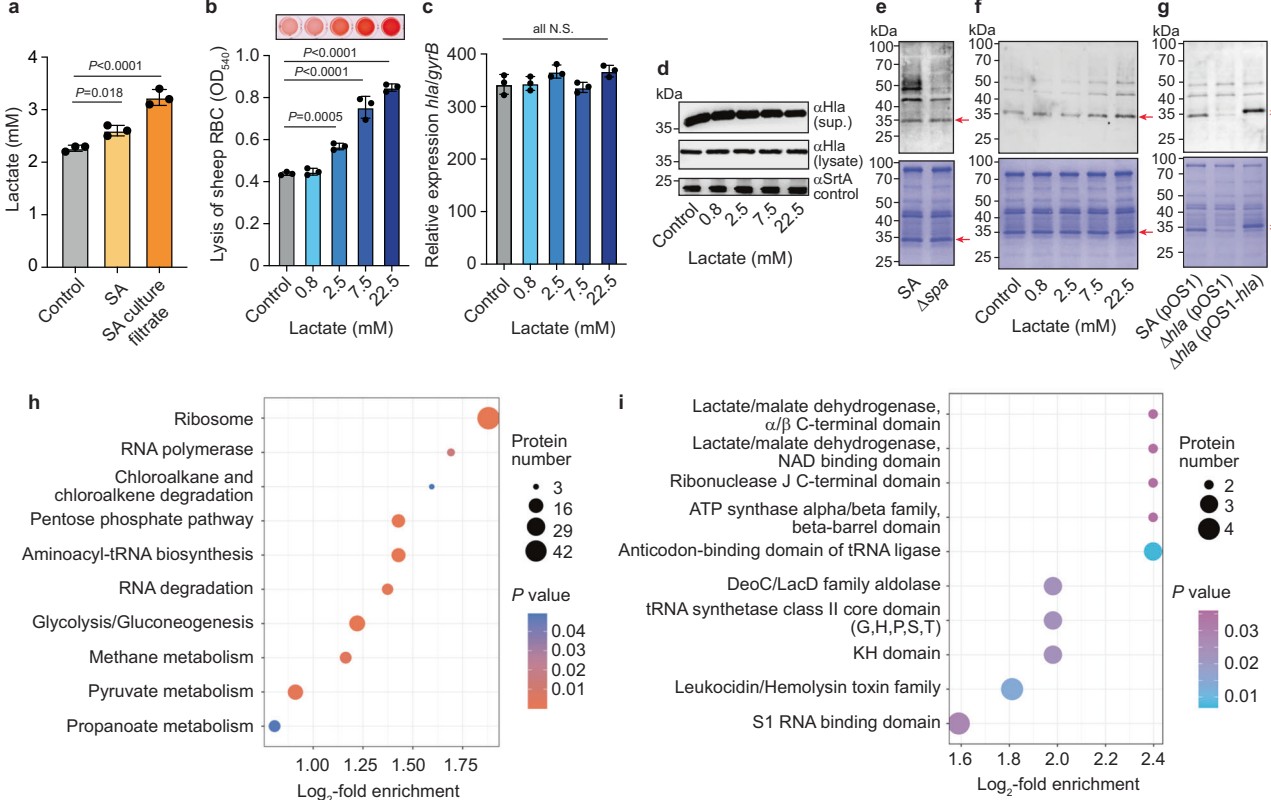

**Fig. 1 | Protein lactylation occurs in *S. aureus* and particularly affects alpha-toxin.** **a** Lactate levels released by A549 cells after incubation with live *S. aureus* (SA, MOI = 5:1) or *S. aureus* stationary-phase culture filtrate (1:50 dilution) for 5 h (n = 3/group). **b** Hemolysis exerted by *S. aureus* culture filtrates containing different concentrations of sodium lactate. Culture filtrates (1:50 dilution) were incubated with RBCs (5% v/v in PBS) for 30 min at room temperature. n = 3/group. **c** Transcript levels of *hla* under the same conditions. n = 3/group. **d** Alpha-toxin protein levels in culture filtrate and cell lysate under the same conditions as determined by immunoblotting with αHla antibodies. Antibodies to sortase A were used to measure sortase protein levels as control. **e** Lactylation of secreted proteins in culture filtrates of *S. aureus* and *S. aureus* Δ*spa* determined by Pan-αKla antibody. See Supplementary Fig. 1f for data with the cellular fraction. **f** Lactylation of *S. aureus* Δ*spa* secreted proteins under the same conditions with addition of different levels of

sodium lactate to cultures. See Supplementary Fig. 1g for data with the cellular fraction. **g** Lactylation in alpha-toxin-deleted (Δ*hla*) and *hla*-complemented *S. aureus* Δ*spa* strains under the same conditions. Lactylated cytoplasmic proteins according to LC-MS/MS analysis: Kyoto Encyclopedia of Genes and genomes (KEGG) pathway analysis (**h**) and protein domain analysis according to InterPro (**i**). All detected lactylation positions of virulence factors are shown in Supplementary Table 3. **e**–**g** Coomassie total proteins stains are shown at the bottom as loading controls. Red arrows denote alpha-toxin; orange arrows the slightly larger His-tagged alpha-toxin. Statistical analysis in (**a**–**c**) is by one-way ANOVAs with Dunnett's post-tests versus values in controls, in (**h**, **i**) by two-sided Fisher's exact test versus all identified proteins or groups. n represents the number of biological replicates for all experiments. Error bars show the mean ± SD. N.S., not significant (P ≥ 0.05).

*aureus* and predominantly due to alpha-toxin (Hla)[4]. We found that lactate in the range of concentrations present in infected, inflamed, or cancerous tissue[20] significantly increased hemolytic capacity of *S. aureus* (Fig. 1b).

Regulatory effects of environmental conditions on virulence-related phenotypes such as cytolysis are commonly based on altered levels of transcription, or more rarely, post-transcriptional or post-translational events, such as secretion, that lead to changed levels of toxin transcript or secreted protein. To our surprise, the increased cytolytic capacity that we detected in lactate-exposed *S. aureus* was not due to either a change in Hla transcript or secreted protein levels (Fig. 1c, d), and we also ruled out that it was due to a lactate-induced pH change (Supplementary Fig. 1c).

Prompted by recent reports on protein lactylation in bacteria[21,22], we asked whether protein lactylation occurs in *S. aureus* and may underlie our findings of lactate-induced cytolysis. We found that specific *S. aureus* proteins in the lysate and in an even more specific fashion, the culture filtrate, reacted with an antibody that binds to lactylated lysines in proteins (Pan-αKla) (Fig. 1e). This was observed in several strains of the ST398 background and in strain USA300 LAC, indicating that the phenomenon was not strain-specific (Supplementary Fig. 1d, e). Removing the non-specific antibody reaction that is due to protein A by creating an isogenic Δ*spa* mutant, the specificity of lactylation could be seen even more clearly (Fig. 1e, Supplementary Fig. 1f). Furthermore, lactylation intensity increased in a dose-dependent fashion with increasing addition of lactate (Fig. 1f, Supplementary Fig. 1g). Notably, among the secreted proteins, the most intense reaction with the Pan-αKla antibody was observed with alpha-toxin, as confirmed by analyzing an *hla* deletion and *hla*-complemented mutant (Fig. 1f, g).

To gain a more encompassing understanding of protein lactylation in *S. aureus*, we performed liquid chromatography tandem mass spectrometry (LC-MS/MS) with *S. aureus* cell lysates after a pull-down with Pan-αKla to decipher the *S. aureus* lactylome (Fig. 1h, i, Supplementary Fig. 2a). These analyses further confirmed disproportionate (i.e., protein-specific) lactylation, since we found specific protein classes, e.g., ribosomal proteins, to be more often subject to lactylation than others. When grouping for protein domains, it was interesting that members of the leucocidin/hemolysin family[28] (specifically, alpha-toxin, gamma-toxin, and LukAB) were among the most represented and those called with the highest confidence, even though most molecules of these secreted proteins should not be present in the lysate (other than as non-secreted preforms or contaminants) (Fig. 1i), which is in accordance with the finding that alpha-toxin was the most pronounced secreted lactylated protein.

## One specific lactylated lysine is responsible for lactylation-dependent cytolysis in alpha-toxin

We then asked which of the several lysines in alpha-toxin that the LC-MS/MS analysis revealed to be subject to lactylation (Supplementary Fig. 2b) is responsible for affecting alpha-toxin's cytolytic capacity. To that end, we expressed His-tagged wild-type alpha-toxin (Hla) or lysine to arginine (K to R) mutants in the respective positions (K63R, K84R, K189R, K266R) in an Hla-deficient (Δ*hla*) background (Supplementary Fig. 3). Only expression of K84R led to significantly decreased cytolysis of RBCs by culture filtrates as compared to wild-type Hla (Fig. 2a). Furthermore, we constructed a K84R genomic mutant of *hla* (SA-*K84R*), which had significantly reduced hemolytic capacity that could be overcome by re-introducing wild-type *hla* (Fig. 2b). Moreover, cytolytic capacity was strongly and significantly reduced only in purified recombinant Hla K84R among all purified wild-type or mutant Hla proteins (Fig. 2c). Finally, lactate dose-dependent increase of cytolysis of RBCs and epithelial A549 cells was absent in purified Hla K84R (Fig. 2d, e). These results indicated that among several lysines of Hla that are subject to lactylation, K84 is specifically

associated with an impact of lactate-concentration-mediated alteration of Hla activity via PTM by lactate.

## Two lactylases are critical for alpha-toxin lactylation-mediated cytolysis

Next, we sought to identify the lactylase enzyme(s) that are responsible for the observed lactylation-dependent phenotypes. To that end, we screened the *S. aureus* ST398 genome for genes potentially related to lysine lactylation and found six genes annotated as encoding N-acetyl transferases. We produced single-gene deletion mutants of all these genes and analyzed them for an impact on lactylation patterns. However, we did not detect any such impact (Supplementary Fig. 4a, b). Hypothesizing that the failure to detect such changes with single gene deletion mutants may be due to functional redundancy, we then over-expressed all six genes and again analyzed lactylation. This analysis pointed to three genes affecting lactylation (*SAPIG0543*, *SAPIG1173*, and *SAPIG2573*) (Fig. 3a, b). Using an in vitro lactylation assay with purified recombinant lactylases, we could show that all these three enzymes were able to lactylate the lysine in a peptide representing the K84-surrounding region of alpha-toxin (Fig. 3c; Supplementary Fig. 4f). Over-expression of the *SAPIG1173* and *SAPIG2573* enzymes led to significantly increased hemolysis and cytolytic activity toward A549 cells in culture filtrates (Fig. 3d, e) and hemolysis by purified Hla (Supplementary Fig. 4g) from those strains. Hla K84R did not exhibit hemolytic activity (Supplementary Fig. 4g). Compared with the SAPIG1173 and SAPIG2573 enzymes, SAPIG0543 only had a weak effect on the hemolytic activity of alpha-toxin (Fig. 3d, e, Supplementary Fig. 4g). To further confirm these results, we constructed single gene deletion mutants in all three genes in question, in addition to a double mutant in the *SAPIG1173* and *SAPIG2573* genes, and a mutant in all three genes (Supplementary Fig. 4c–e). Analyses of culture filtrates, purified Hla proteins, and the Δ*SAPIG1173* + *SAPIG2573* double mutant with and without addition of lactate and in RBCs as well as A549 cells corroborated that *SAPIG1173* and *SAPIG2573* are the lactylases primarily responsible for lactylation-dependent Hla-mediated cytolytic phenotypes in a fashion that is dependent on external lactate concentration (Fig. 3f–i, Supplementary Fig. 4h, i). Analysis of hemolytic activity was also again performed with Hla K84R, which was strongly reduced, further corroborating the specific role of K84R (Supplementary Fig. 4h). We also complemented the Δ*SAPIG1173* + *SAPIG2573* mutant separately with plasmids expressing either the SAPIG1173 or SAPIG2573 enzymes, which restored wild-type level cytolytic capacities of culture filtrates and purified Hla (Fig. 3j, k, Supplementary Fig. 4j, k). Finally, we performed an in vitro lactylation assay with the K84-surrounding peptide and purified recombinant SAPIG1173 or SAPIG2573 enzymes, using LC-MS/MS detection of lactylation, which showed activity with both enzymes (Fig. 3l, Supplementary Fig. 4l). These results pinpoint the functional lactylation of alpha-toxin that leads to increased cytolytic activity specifically to two enzymes, SAPIG1173 and SAPIG2573.

## Lactylation affects receptor-independent alpha-toxin membrane association

Many details of the structure-function relationship of alpha-toxin and its oligomerization to a functional heptameric pore remain poorly understood[4]. While a detailed investigation of how lactylation affects alpha-toxin activity on a molecular level is thus beyond the scope of the present study, we still aimed to gain preliminary insight into which categorical aspects of that process may be impacted. We first noted that the K84 residue in an alpha-toxin heptameric pore is located in the cap region, thus likely not being directly involved in membrane integration (Fig. 4a).

One categorical aspect of alpha-toxin pore formation is receptor-mediated versus receptor-independent association of the alpha-toxin

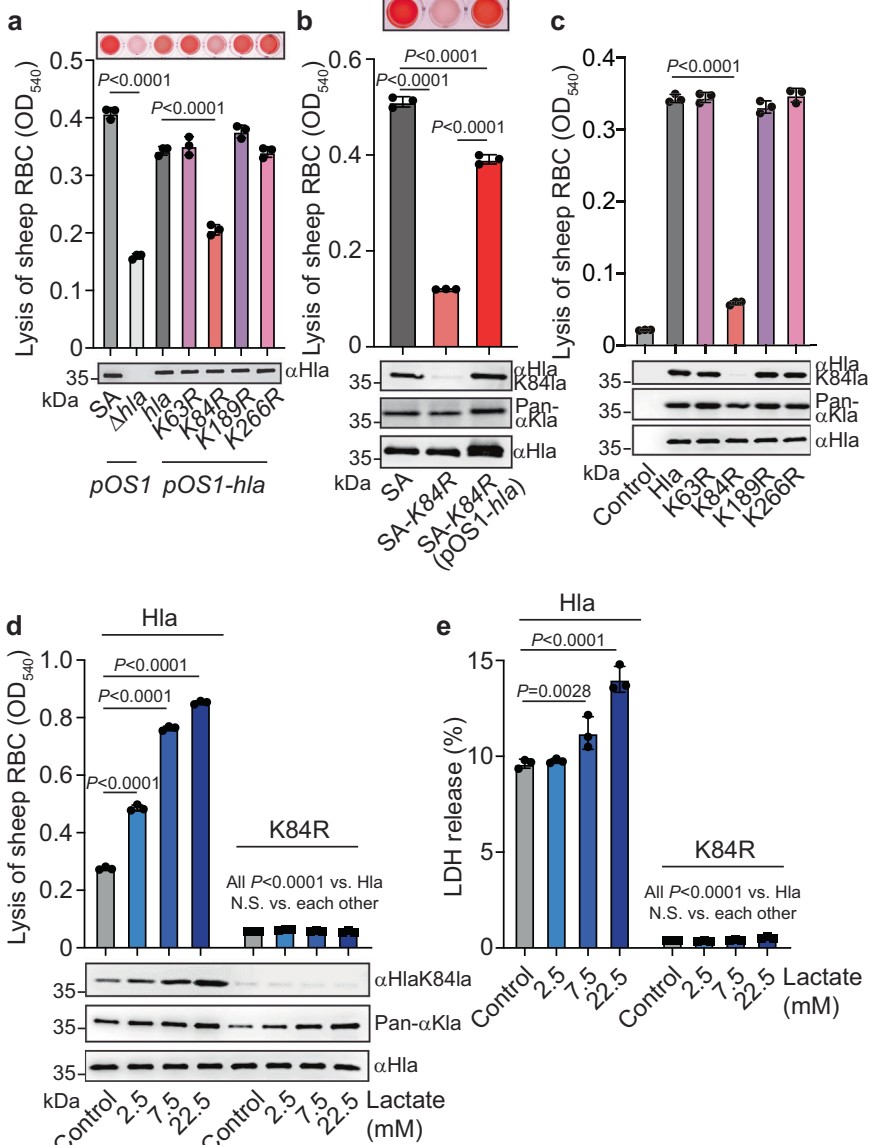

**Fig. 2 | One specific lactylated lysine is responsible for lactylation-dependent cytolysis in alpha-toxin. a** Cytolytic activity (as determined by lysis of RBCs) of culture filtrates (1:50 dilution) from *S. aureus* strains expressing wild-type (*hla*) or different single-site amino acid substitutions of alpha-toxin and control strains. Immunoblot controls of expression levels are shown at the bottom. **b** Cytolytic activity toward RBCs of culture filtrates (1:50 dilution) from *S. aureus* wild-type and an isogenic *hla* K84R genomic single-site substitution strain (SA-*K84R*), and wild-type *hla*-complemented strain. **a**, **b** Representative images are shown at the top. **c** Cytolytic activity toward RBCs of purified wild-type and single-site amino acid substitutions of alpha-toxin. Lactate-dependent cytolysis of RBCs (**d**) or A549 cells (**e**) by purified wild-type Hla or Hla K84R. **b**–**d** Reactions with antibodies specific for alpha-toxin (αHla) or K84-lactylated alpha-toxin (αHlaK84la), and antibodies reacting with all lysine-lactylated proteins (Pan-αKla) are shown below graphs. **a**–**e** *n* = 3/group (biological replicates). Statistical analysis is by one-way ANOVAs with Tukey's post-tests. Error bars show the mean ± SD.

heptamer with the target membrane. The former is mostly dependent on the disintegrin and metalloprotease ADAM-10[29,30], which is believed to interact initially with an alpha-toxin monomer, and caveolin-1[30,31], which is believed to form an anchor for alpha-toxin parts during membrane insertion[32]. This occurs already at low concentrations, while receptor-independent pore formation is affected by membrane lipid composition and occurs only at higher (>1 μM) concentrations of alpha-toxin[33]. According to established models, steps preceding the formation of a stable alpha-toxin heptameric pore[34] in the target membrane include binding of a monomer to the membrane, oligomerization to a membrane-associated heptameric pre-pore that undergoes conformation changes leading to SDS-stability, and finally considerable structural changes that are associated with membrane integration[35] (Fig. 4b). While the latter transition is well

investigated, the structural prerequisites in the alpha-toxin molecule for monomer membrane association and oligomerization are poorly understood.

In a pull-down assay with purified Hla and A549 cell lysate, we did not detect differences in the amount of ADAM-10 or caveolin-1 pulled down by wild-type versus K84R alpha-toxin, indicating that receptor interaction is not affected by changes at position 84 in the alpha-toxin molecule (Fig. 4c). These results suggest that alpha-toxin lactylation affects receptor-independent membrane association of alpha-toxin.

**Alpha-toxin lactylation affects in vivo pathogenesis**

We then analyzed whether lactylation impacts pathogenesis in vivo, using mouse models of skin and lung infection. Mice infected with the Δ*SAPIG1173* + *SAPIG2573* strain showed significantly reduced abscess

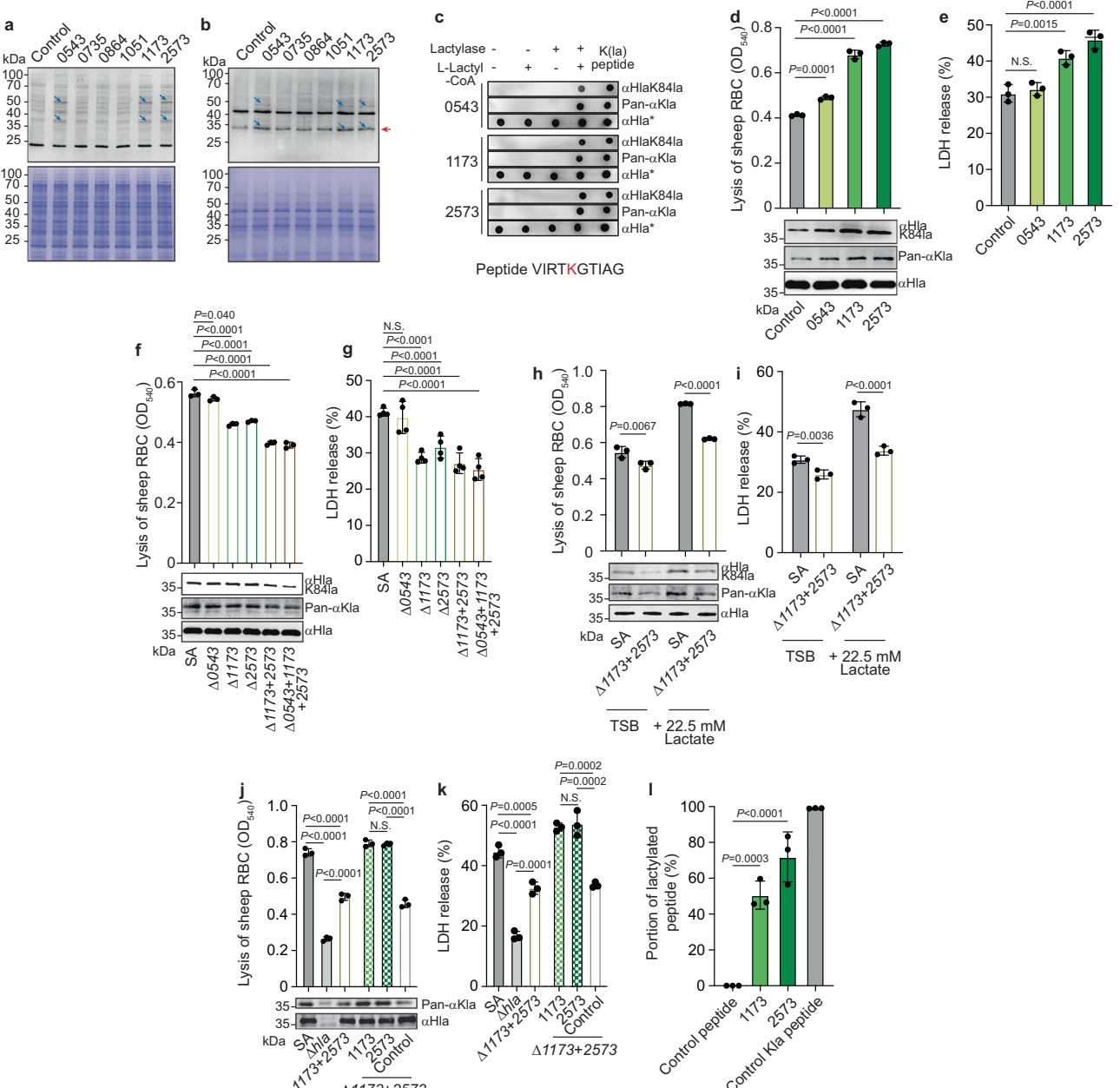

**Fig. 3 | Two lactylases are critical for alpha-toxin lactylation-mediated cytolysis.** Lactylation of proteins in cell lysates (**a**) or culture filtrates (**b**) of *S. aureus* Δ*spa* strains over-expressing specific potential lactylases (*n* = 3 independent experiments). Gene names are according to a genome-sequenced ST398 strain[43] ("0543" for SAPIG0543, etc.). The alpha-toxin band is marked by a red arrow. Blue arrows show examples of differentially expressed proteins. Coomassie whole-protein stains are shown at the bottom as controls. **c** In vitro lactylase activity assay. Purified recombinant (His-tagged) lactylases were incubated with a peptide representing the region surrounding position 84 of Hla. Analysis was by immune dot blot using antibodies developed to react only with K84-lactylated Hla (αHlaK84la, developed against the lactylated form of this peptide and affinity-purified) or control antibodies developed against the corresponding unmodified peptide, reacting with both modified and unmodified Hla (αHla*), and antibodies reacting with all lysine-lactylated proteins (Pan-αKla). See Supplementary Fig. 3 for analysis of antibody specificity. The K84-lactylated peptide was used a positive

control. Cytolytic activity toward RBCs (**d**) or A549 cells (**e**) of culture filtrates of the lactylase expression strains. **f–i** Data obtained using lactylase single or multiple deletion strains. Cytolytic activity of culture filtrates toward RBCs (**f**) and A549 cells (**g**). **h**, **i** Cytolytic activity data obtained with the double Δ*1173* + *2573* mutant without or with addition of sodium lactate. **j**, **k** Cytolytic activities obtained by genetic complementation of the Δ*1173* + *2573* mutant by single expression of lactylase enzymes. (**l**), In-vitro enzyme/substrate conversion assay using purified 1173 and 2573 lactylases and the peptide shown in (**c**). Detailed source data for this assay are available in Supplementary Data 1. **d–l** *n* = 3/group except (**g**), *n* = 4/group. n represents the number of biological replicates for all experiments. Reactions with αHla, αHlaK84la, and Pan-αKla antibodies are shown at the bottom of graphs. Statistical analysis is by 1-way ANOVA and Tukey's or Dunnett's post-tests via controls, as appropriate. In (**j**, **k**) only the values obtained among plasmid or non-plasmid-containing groups were compared. Error bars show the mean ± SD.

sizes in the skin infection model (Fig. 5a) and significantly reduced mortality and bacterial burdens in the lung infection model (Fig. 5b, c), as compared to those infected with wild-type *S. aureus*. Analysis of bronchoalveolar lavage fluid (BALF) also demonstrated reduced

presence of leukocytes and inflammatory cytokines (IL-6, TNF-α, IL-1β) (Fig. 5d–i). Histological examination of lung tissue showed pronounced neutrophil influx, alveolar atelectasis, and perivascular edema in mice infected with wild-type *S. aureus*, while these effects

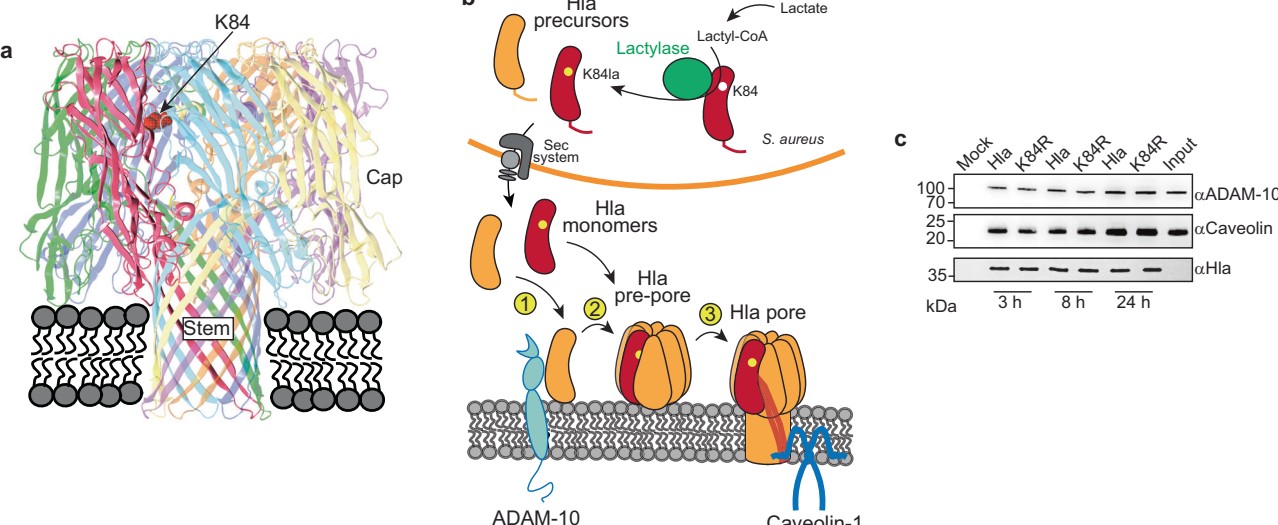

**Fig. 4 | Lactylation affects receptor-independent alpha-toxin membrane association. a** Model of the alpha-toxin pore according to the structure reported by Song et al.[34], rendered using Lasergene 17 Protean 3D. The position of K84 is highlighted using space fill presentation of the lysine side chain within one monomer shown in red. Note we counted amino acids from the start of the Hla precursor peptide (not the mature toxin) in this study. **b** Model of alpha-toxin (Hla) lactylation and steps leading to pore formation: (1) Attachment of Hla monomer to membrane involving ADAM-10 receptor, (2) oligomerization on the membrane, which includes structural changes leading to SDS stability, (3) pore formation involving considerable structural changes including stem formation and anchoring to caveolin-1. **c** Binding ability of alpha-toxin to ADAM−10 and caveolin-1 (*n* = 3 independent experiments). Purified His-tagged wild-type or K84R alpha-toxin were incubated with A549 cell lysate for the indicated times. The binding proteins were collected via His-tag pull-down and detected by immunoblot. "Input": A549 cell lysate only; "Mock": A549 cell lysate incubated with equal amount of PBS for 24 h.

were virtually absent in mice infected with the ΔSAPIG1173 + SAPIG2573 strain (Fig. 5j, k). These results showed a significant impact of protein lactylation on *S. aureus* in vivo virulence.

Because it is impossible to genetically create a strain in which lactylation exclusively of alpha-toxin is abrogated, we used alternative approaches to link the observed effects of lactylation on *S. aureus* pathogenesis specifically to alpha-toxin. First, we used the SA-*K84R* mutant strain. Mice infected with that strain revealed significantly reduced abscess sizes in the skin infection model and significantly reduced mortality, bacterial burdens, and leukocyte influx in the lung infection model (Fig. 6a–e). While these results establish the importance of the K84 residue in alpha-toxin for in vivo virulence, they do not unambiguously link the effects to lactylation as the K to R mutation may impact alpha-toxin function independently of lactylation. Therefore, we constructed a strain in which the *hla* gene is deleted in addition to the *SAPIG1173* and *SAPIG2573* genes (Δ*hla* + *SAPIG1173* + *SAPIG2573*) and compared it in the two infection models to effects exerted by the Δ*SAPIG1173* + *SAPIG2573* strain. The infecting CFU was increased in this experiment in both groups to compensate for the decreased virulence of the Δ*hla* background of the control strain. There were no significant differences between mice infected with the two strains in any of the parameters for which we had found significant differences comparing mice infected with wild-type *S. aureus* versus mice infected with the Δ*SAPIG1173* + *SAPIG2573* strain (Fig. 6f–p). These results show that the virulence defect of the Δ*SAPIG1173* + *SAPIG2573* strain is predominantly due to alpha-toxin, indicating that lactylation specifically of alpha-toxin contributes to *S. aureus* in vivo virulence.

## Discussion

In this study, we provide evidence for PTM of *S. aureus* proteins by lactylation. Notably, lactylation occurred in a non-random fashion, and among secreted *S. aureus* proteins, the most strongly lactylated protein was alpha-toxin, a well-studied, key mediator of *S. aureus* virulence[4,36]. We also identified the lactylase enzymes that are responsible for alpha-toxin lactylation, linked lactylation specifically of alpha-toxin to phenotypes important for pathogenesis as well as directly to in vivo virulence in animal infection models, and pinpointed the lactylated residue in alpha-toxin that is critical in that context to lysine in position 84.

While the only recently discovered protein modification by lactylation has so far been primarily been found and studied in eukaryotes[19,20], our results emphasize that it also has important functions in bacteria and adds the human pathogen *S. aureus* to the hitherto short list of bacteria where this type of PTM has been described[21,22]. Our finding that protein lactylation in *S. aureus* occurs in a non-random fashion and especially affects alpha-toxin is surprising, as alpha-toxin is a well-studied protein toxin where this has not been previously noted[4]. It was even more unexpected that this modification was necessary for alpha-toxin to achieve maximal cytolytic activity, as the dependence of bacterial toxin activity on PTM has only been reported in very rare cases[17,18].

Why lactylation of alpha-toxin has not been previously noted maybe because it represents a small change in mass and occurs only in a fraction of alpha-toxin molecules. The latter is in accordance with most PTMs, which are commonly substoichiometric[37]. While the exact percentage is hard to determine based on MS analyses, we found that addition of lactate substantially increases that fraction. Notably, this was observed at concentrations of lactate that are also observed during infection.

How exactly lactylation of alpha-toxin at position K84 promotes activity remains to be studied in detail. Our preliminary experiments rule out receptor interaction and point to oligomerization as critical step. Further investigation into this direction will be challenging, as the alpha-toxin oligomerization step is generally not well understood on a molecular level[35]. That histidine residues in the N-terminal region of alpha-toxin have previously been implicated in oligomerization is in accordance with the idea of an impact of the extended N-terminal region, which includes K84, in that process[38]. Moreover, previous studies have shown that the formation of a functional alpha-toxin pore

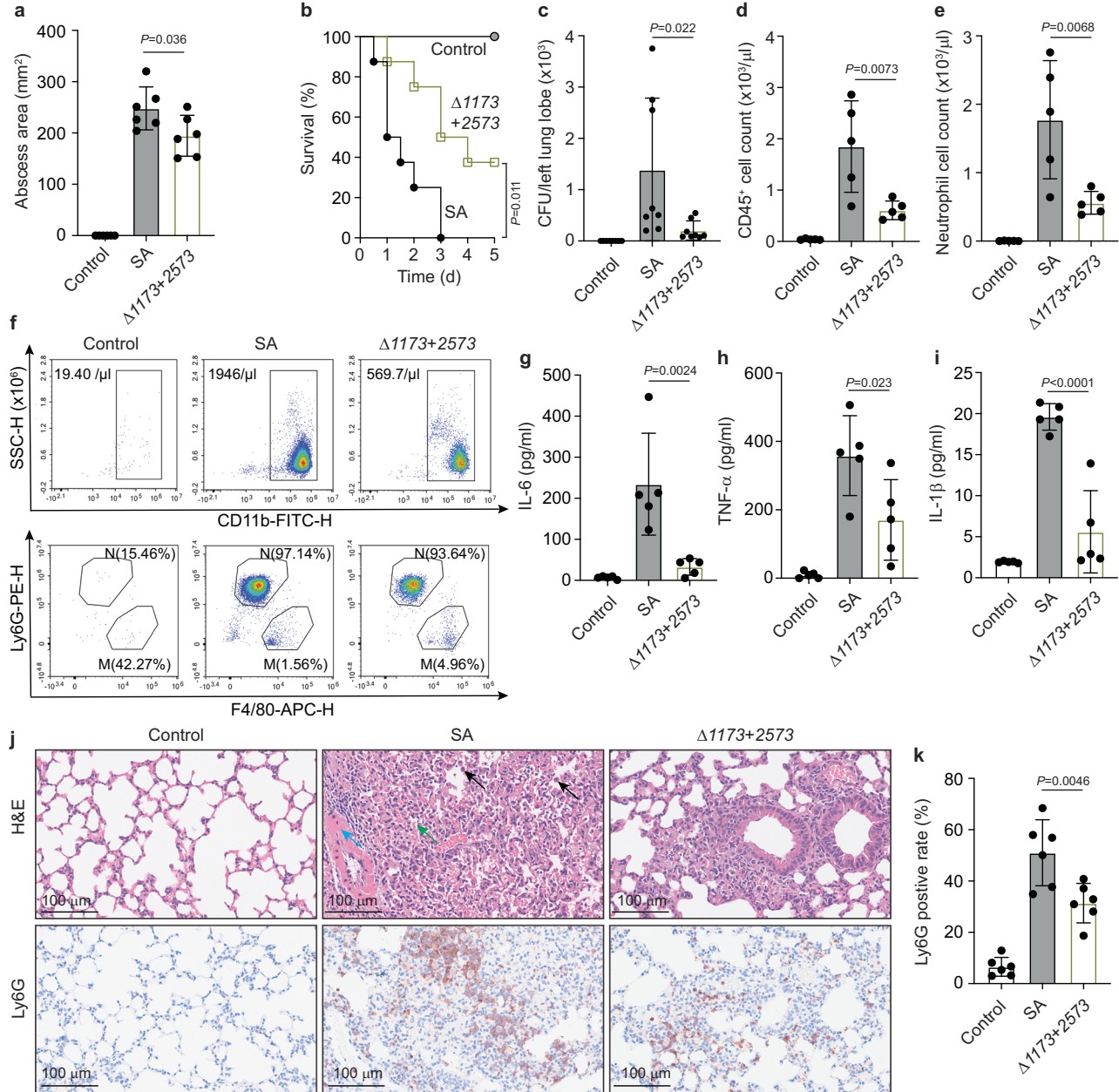

**Fig. 5 | Lactylation affects *S. aureus* infection. a** Abscess model. Mice received 2.5 × 10⁷ CFU in 100 µl PBS of the indicated strains or PBS alone by intradermal infection and abscess size was measured 24 h thereafter. *n* = 6/group. **b**–**k** Lung infection model. Mice received 2 × 10⁹ CFU (lethal dose, (**b**) or 2 × 10⁸ CFU (non-lethal dose, (**c**–**k**) in 40 µl PBS of the indicated strains or PBS alone by intranasal instillation. **b** Survival curve. *n* = 8/group. In a different group of mice receiving the non-lethal dose, the bacterial load (**c**) was determined after euthanasia at 48 h (*n* = 8/group), and the BALF was examined (*n* = 5/group) for the presence of lymphocytes (CD45⁺ cells, **d**)) and neutrophils (**e**) using the gating strategy shown in Supplementary Fig. 5. **f** Representative contour plots of CD11b staining showing abundance of myeloid cells, most of which are neutrophils (Ly6G⁺) and

macrophages (F4/80⁺) in the BALF of mice infected with wild-type *S. aureus* (SA) or *Δ1173 + 2573* bacteria. In the top panel, the count of CD11b positive cells, and in the bottom panel, the proportion of neutrophils (N) and macrophages (M) is shown. See gating strategy in Supplementary Fig. 5. **g**–**i** Cytokine concentrations in the BALF (*n* = 5/group). **j, k** Lung tissue was examined by histology (H&E staining, anti-Ly6G immunohistochemistry to detect leukocytes), *n* = 6/group. Black arrows depict leukocyte infiltration, green arrows alveolar atelectasis, and blue arrows perivascular edema. *n* represents the number of biological replicates for all experiments. Statistical analysis is by Mantel-Cox log-rank test for the survival curve and elsewhere by one-way ANOVAs with Tukey's post-tests. Error bars show the mean ± SD.

can be dependent on single amino acid exchanges in only one of the seven molecules that form the heptameric pore[39,40], which would be in accordance with functional, substoichiometric lactylation at K84 and the alteration of alpha-toxin cytolytic functionality by the lactate-dependent increase in lactylation that we observed.

In conclusion, post-translational protein modification by lactate, particularly of alpha-toxin, affects *S. aureus* virulence and is dependent on high lactate concentration, which occurs during infection. This

establishes a link between the environmental conditions during infection and virulence that is accomplished not via the conventional way involving bacterial sensors and regulatory proteins, such as two-component systems[41], but in a previously unknown fashion by post-translational modification of a key toxin. Furthermore, our study suggests that owing to their role in virulence that we established, lactylases in *S. aureus* and potentially other pathogens may be targets for anti-virulence drug development.

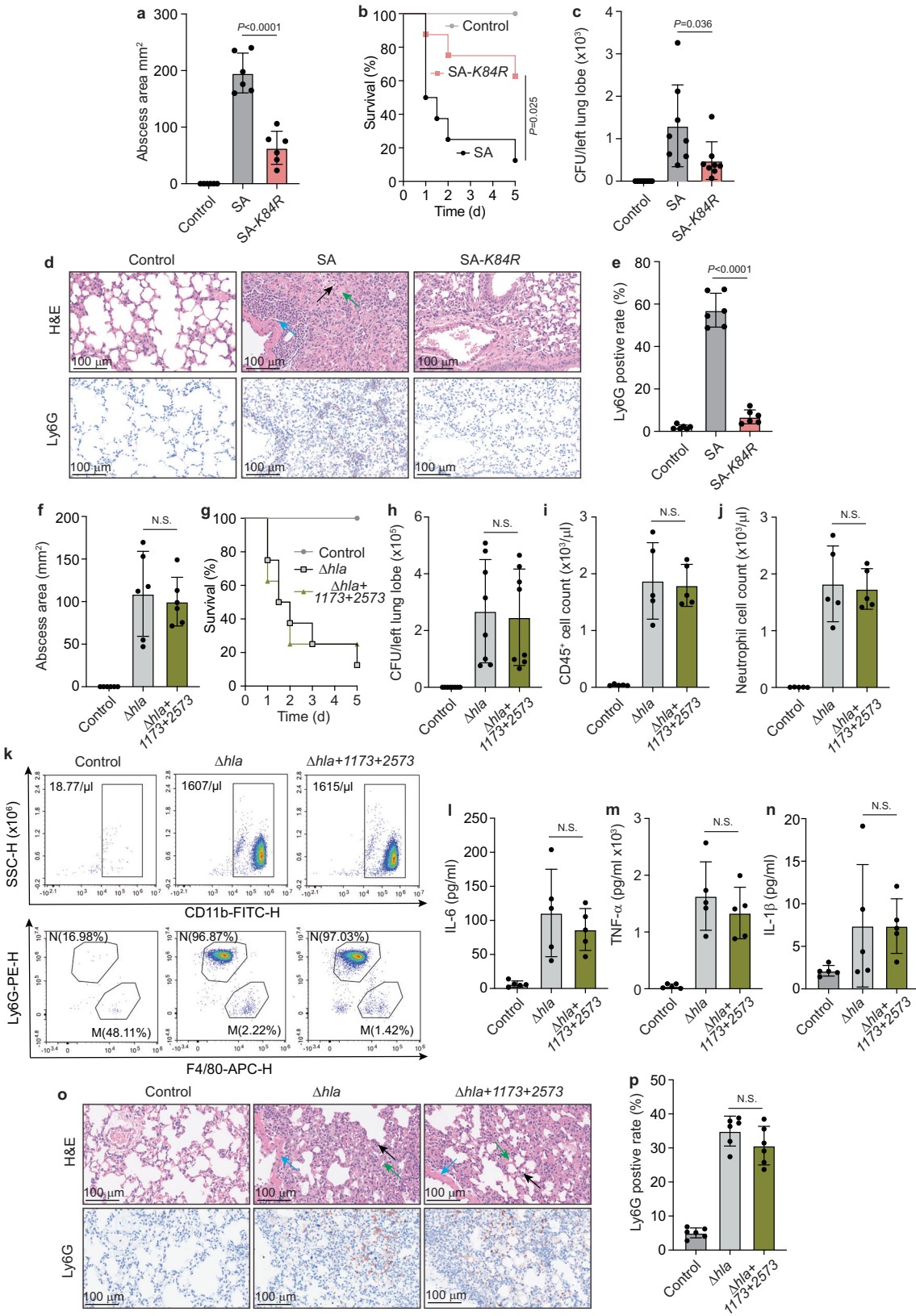

## Methods

### Mice

All animal experiments were performed in accordance with the laboratory animal care and use guidelines of the Chinese Association for Laboratory Animal Sciences (CALAS). Approval was obtained from the Ethics Committee for Experimental Animal Welfare at Renji Hospital, School of Medicine, Shanghai Jiao Tong University, Shanghai (RJ2023-087A). BALB/c female mice were purchased from Shanghai JSJ Laboratory Animal Co, Ltd. Animals were housed at 19–26 °C at a humidity of 40–70% and with a light/dark cycle of 12 h/12 h in-house under SPF conditions. Age- and littermate-matched mice were randomly assigned into treatment groups in each experiment. Female

**Fig. 6 | Lactylation of alpha-toxin dominates in vivo pathogenesis effects of protein lactylation in *S. aureus*. a–e** Mouse models using SA-*K84R* strain. **a** Abscess model. Mice received $2.5 \times 10^7$ CFU in 100 µl PBS of the indicated strains or PBS alone by intradermal infection and abscess size was measured 24 h thereafter. $n = 6$/group. **b–e** Lung infection model. Mice received $2 \times 10^9$ CFU (lethal dose, (**b**)) or $2 \times 10^8$ CFU (non-lethal dose, (**c–e**)) in 40 µl PBS of the indicated strains or PBS alone by intranasal instillation. **b** Survival curve. $n = 8$/group. In a different group of mice receiving the non-lethal dose, the bacterial load (**c**) was determined after euthanasia at 48 h ($n = 8$/group). Lung tissue was examined by histology (**d**) using H&E staining and anti-Ly6G immunohistochemistry to detect leukocytes (**e**), $n = 6$/group. **f–p** Mouse models using the *Δhla* background. **f** Abscess model. Mice received $1 \times 10^8$ CFU in 100 µl PBS of the indicated strains or PBS alone by intradermal infection and abscess size was measured 24 h thereafter. $n = 6$/group. **g–p** Lung infection model. Mice received $8 \times 10^9$ CFU (lethal dose, (**g**)) or $8 \times 10^8$ CFU (non-lethal dose, (**h–p**)) in 40 µl PBS of the indicated strains or PBS alone by intranasal instillation. **g** Survival curve. $n = 8$/group. In a different group of mice receiving the non-lethal dose, the bacterial load (**h**) was determined ($n = 8$/group) after euthanasia at 48 h and the BALF was examined ($n = 5$/group) for the presence of lymphocytes (CD45$^+$ cells, (**i**)), neutrophils (**j, k**), and cytokines (**l–n**). **k** Representative contour plots of CD11b staining showing abundance of myeloid cells, most of which are neutrophils (Ly6G$^+$) and macrophages (F4/80$^+$) in the BALF of mice infected with *Δhla* or *Δhla* + *1173* + *2573* bacteria. In the top panel, the count of CD11b positive cells, and in the bottom panel, the proportion of neutrophils (N) and macrophages (M) is shown. See gating strategy in Supplementary Fig. 5. **o, p** Lung tissue was examined by histology (H&E staining, anti-Ly6G immunohistochemistry to detect leukocytes), $n = 6$/group. **d, o** Black arrows depict leukocyte infiltration, green arrows alveolar atelectasis, and blue arrows perivascular edema. n represents the number of biological replicates for all experiments. Statistical analysis is by Mantel-Cox log-rank test for the survival curves and elsewhere by one-way ANOVAs with Tukey's post-tests. Error bars show the mean ± SD.

---

mice were used as they are more docile and less difficult to handle. This is common in the field to monitor *S. aureus* experimental lung and skin infections, which are not known to be influenced by gender.

## Bacterial isolates and growth conditions

Wild-type *S. aureus* strains were isolated from adult patients of Ren Ji Hospital affiliated with Shanghai Jiaotong University. All participants or their legal guardians have provided written informed consent to take part in the study. This study was approved by the ethics committee of Renji Hospital, School of Medicine, Shanghai Jiao Tong University, Shanghai (KY2023-060-C).

Bacterial strains used in this study are listed in Supplementary Table 1. Unless otherwise noted, *S. aureus* strains were grown in tryptic soy broth (TSB), and *E. coli* strains were grown in Luria-Bertani broth (LB) at 37 °C. For plasmid maintenance, antibiotics were used at the following concentrations: ampicillin, 100 µg/ml; erythromycin, 10 µg/ml; chloramphenicol, 10 µg/ml; kanamycin, 50 µg/ml.

## Cell line and culture conditions

A549 cells (catalog number SCSP-503), sourced from the Cell Bank of Shanghai Institutes of Biological Sciences, Chinese Academy of Sciences, were cultured in RPMI 1640 medium supplemented with 10% fetal bovine serum (Gibco), as well as penicillin (100 U/ml) and streptomycin (0.1 mg/ml) at 37 °C. The A549 cell line was authenticated by short tandem repeat (STR) profiling. Amelogenin: X,Y; CSF1PO: 10,12; D13S317: 11; D16S539: 11,12; D5S818: 11; D7S820: 8,11; THO1: 8,9.3; TPOX: 8,11; vWA: 14.

## Construction of plasmids for gene expression

To construct the *hla* expression plasmid for genetic complementation, the *hla* gene (including its promoter region and a region encoding a C-terminal His-tag) was PCR-amplified, introducing EcoRI and BamHI sites. The *pOS1* plasmid and the amplified fragment were digested using the restriction enzymes EcoRI and BamHI. The digested fragment was ligated into *pOS1* and transformed into *E. coli* DH5α; then the resulting plasmid *pOS1-hla* was purified from *E. coli* and transformed by electroporation into *S. aureus*.

To construct the lactylase overexpression plasmids, the PCR-amplified fragment and plasmid *pYJ335* were subjected to digestion with restriction enzymes EcoRV and KpnI. Then, the digested fragment was ligated into *pYJ335* and transformed into *E. coli* DH5α. The resulting plasmid *pYJ335-lactylase* was subsequently transformed by electroporation into *S. aureus*. Of note, the lactylase gene is positioned downstream of the tetracycline-inducible *xyl/tetO* promoter in *pYJ335*, thereby enabling its induction by tetracycline (50 ng/ml).

For the in vitro expression of lactylase, genes *SAPIG0543*, *SAPIG1173*, and *SAPIG2573* were amplified by PCR. The PCR fragments and plasmid *pET28a* were digested with BamHI and EcoRI, ligated, and

transformed into *E. coli* DH5α. Then, resulting *pET28a*-lactylase plasmids were transformed into *E. coli* BL21 (DE3) for protein expression. To induce protein expression, a final concentration of 0.5 mM isopropyl-β-D-thiogalactopyranoside (IPTG) was used.

All plasmids used in this study are listed in Supplementary Table 1.

## Construction of gene mutants

A ligation-independent cloning method was used for gene deletion. The homologous recombination procedure was performed using plasmid *pKOR1* as described[42]. See Supplementary Table 2 for all primers used in this study.

The KOD-Plus-Mutagenesis Kit (TOYOBO) was used to induce point mutations. The *pOS1* plasmid carrying a wild-type *hla* gene was used as a template for inverse PCR using the mutation primers listed in Supplementary Table 2. The PCR products were digested with DpnI and self-ligation was performed by a reaction with T4 polynucleotide kinase and ligase. Then, the self-ligated PCR products were transformed into *E. coli* DH5α (Thermo Scientific) and PCR-amplified sequences were verified by DNA sequencing.

## Lysis of erythrocytes

Bacterial overnight cultures were diluted 1:100 into fresh TSB, incubated at 37 °C with shaking for 15 h, and culture filtrates were obtained by centrifugation (4 °C, $6000 \times g$, 10 min). The culture filtrates (1:50 dilution) or purified α-hemolysin (final concentration, 10 µg/ml) were incubated with sheep RBCs (Sbjbio, Nanjing, Co., Ltd; 5% v/v in PBS) for 30 min at 25 °C. After incubation, the samples were centrifuged at $1500 \times g$ for 10 min, and the supernatant was transferred into a 96-well flat-bottom plate. The hemolytic ability of samples was evaluated by measuring the absorbance at 540 nm using a BioTek Synergy2 microplate reader.

## Quantitative reverse transcription (qRT)-PCR

Total RNA of *S. aureus* was extracted using an RNease Kit (Qiagen) and complementary DNA was synthesized from the RNA using a Quanti-Tect reverse transcription system (Qiagen). The resulting complementary DNA samples were amplified using a QuantiTect SYBR green PCR kit (Qiagen) and qRT-PCR was performed on a 7500 Sequence Detector system (Applied Biosystems) in MicroAmp Optical 96-well reaction plates.

## *S. aureus* stimulation of lactate production in A549 cells

Cultures inoculated 1:100 from pre-cultures grown overnight were incubated for 8 h at 37 °C with shaking. A549 cells were incubated with bacterial culture filtrates (1:50 dilution) or live *S. aureus* (MOI = 5:1 *S. aureus* to A549 cells) at 37 °C for 5 h. After incubation, samples were centrifuged at $500 \times g$ for 10 min to collect cell supernatant. A Lactate Assay Kit (Sigma) was used to measure the lactate released by the A549 cells.

## LC-MS/MS analysis

Bacteria were harvested from 10-h cultures (grown in TSB at 37 °C with shaking), which were inoculated 1:100 from overnight pre-cultures. Lysis buffer (1% Triton X-100, 1% protease inhibitor cocktail, 3 μM trichostatin A, 50 mM nicotinamide) was added and then the bacterial suspension was sonicated (250 W, 3 min, with a cycle of 5 s sonication and 7 s pause on ice) using a high-intensity ultrasonic processor (Scientz). Afterward, samples were centrifuged at $12,000 \times g$ for 10 min at 4 °C to collect the supernatant containing *S. aureus* proteins. The protein concentration was determined using a BCA protein assay kit (Beyotime Biotech Co., Ltd) according to the manufacturer's instructions. Equal amounts of protein from each sample were then subjected to trypsin digestion. To one volume of protein solution, five volumes of acetone were added to precipitate proteins (−20 °C, 2 h). The precipitate was collected by centrifugation ($4500 \times g$, 5 min), washed twice with pre-cooled acetone, and air-dried. Tetraethyl ammonium bicarbonate (TEAB) was added to a final concentration of 200 mM and samples were sonicated as above to homogenize the precipitation. Trypsin was added at a 1:50 trypsin:protein mass ratio and samples were incubated overnight at 37 °C. Then, samples were treated with 5 mM dithiothreitol at a final concentration of 5 mM at 56 °C for 30 min and alkylated with iodoacetamide at a final concentration of 11 mM in the dark at 25 °C for 15 min. The obtained tryptic peptides were dissolved in immunoprecipitation (IP) buffer (50 mM Tris-HCl, 100 mM NaCl, 1 mM EDTA, 0.5% NP-40, pH 8.0), and incubated with anti-L-lactyl-lysine antibody-conjugated agarose beads (PTM-1404, PTM Biolabs) overnight at 4 °C with gentle shaking. Then, the beads were washed four times with IP buffer and twice with ddH2O. The bound peptides were released from the beads by eluting three times with 0.1% trifluoroacetic acid. The eluates were dried and desalted using C18 ZipTips (Millipore) before LC-MS/MS analysis.

For LC-MS/MS, the peptide samples were dissolved in solvent A (0.1% formic acid, 2% acetonitrile in water) and separated with the following gradient to solvent B (0.1% formic acid in acetonitrile) using a NanoElute UHPLC system (Bruker Daltonics) coupled to a timsTOF Pro mass spectrometer (Bruker Daltonics): 6–24% in 40 min, 24–32% in 12 min, 32–80% in 4 min, 80% for 4 min, at a constant flow rate of 450 nl/min. The electrospray voltage applied was 1.70 kV. Precursors and fragments were analyzed at the TOF detector, employing a comprehensive MS/MS scan range from 100 to 1700 m/z. The timsTOF Pro instrument was operated in the parallel accumulation serial fragmentation (PASEF) mode. Precursors with charges state from zero to five were selected for fragmentation, while each cycle encompassed the acquisition of ten PASEF-MS/MS scans. To ensure optimal accuracy, a dynamic exclusion period of 30 s was implemented.

## Database search

Mass spectrometry data were analyzed using Maxquant (version1.6.15.0). The reference database was *S. aureus* ST398 (Genome assembly ASM958v1). Trypsin/P was specified as cleavage enzyme allowing up to four missing cleavages. The minimum length of the peptide segment was set to seven amino acid residues. The maximum number of peptide modifications was set to five. The mass tolerance for precursor ions was set at 20 ppm in the first and 20 ppm in the main search, and the mass tolerance for fragment ions was set at 20 ppm. Carbamidomethyl on cysteine was considered a fixed modification and acetylation on the protein N-terminus and oxidation of methionine were considered variable modifications. The false discovery rate (FDR) was adjusted to <1% and any identified protein had to contain at least one unique peptide.

## KEGG pathway analysis

The Kyoto Encyclopedia of Genes and Genomes (KEGG) database (https://www.kegg.jp/) was used to identify enriched pathways, subject to analysis of statistical significance using a two-tailed Fisher's exact test, to test the enrichment of the modified protein compared to all identified proteins. The pathway with a corrected $P$ value of $P < 0.05$ was considered significant.

## Protein domain analysis

For each protein category, the InterPro database (a resource that provides functional analysis of protein sequences by classifying them into families and predicting the presence of domains and important sites, https://www.ebi.ac.uk/interpro/) was searched and a two-tailed Fisher's exact test was employed to test the enrichment of the modified protein compared to all identified proteins. Protein domains with a corrected $P$ value of $P < 0.05$ were considered significant.

## Gene ontology (GO) analysis

Gene Ontology (GO) annotation of the proteome was derived from the UniProt-GOA database (http://www.ebi.ac.uk/GOA/). If proteins were not annotated by UniProt-GOA database, the InterProScan soft was used to annotate a protein's GO function based on protein sequence alignment. Then, proteins were classified by Gene Ontology annotation based on three categories: biological process, cellular component, and molecular function. For each category, a two-tailed Fisher's exact test was employed to test the enrichment of the modified protein compared to all identified proteins. Results with a corrected $P$ value of $P < 0.05$ were considered significant.

## Purification of recombinant proteins

For the purification of His-tagged alpha-toxin, 10 ml overnight TSB cultures of *S. aureus* carrying the plasmid *pOS1-hla* were diluted 1:100 into 200 ml fresh TSB medium supplemented with 10 μg/ml chloramphenicol and grown at 37 °C with shaking (200 rpm) for 15 h. In some experiments, sodium lactate (Sigma) was added to the TSB medium at the given concentrations. To maintain the pH, HEPES solution (pH 7.5) was added to the culture medium to a final concentration of 30 mM. After centrifugation, the culture supernatant was collected and filtered using syringe filters (0.45 μm, Millipore). Alpha-toxin in the supernatant was collected using polypropylene columns (Beyotime Biotech Co., Ltd) self-packed with Ni-NTA agarose (Qiagen), affinity chromatography was performed using gravity flow, and alpha-toxin was eluted with 250 mM imidazole elution buffer. Finally, imidazole in the sample was removed by ultrafiltration centrifugation using Amicon Ultra-15 10 K centrifugal filter devices (Millipore).

To express and purify putative lactylases, we cloned the corresponding genes into plasmid *pET28a* and transformed the resulting expression constructs into *E. coli* BL21 (DE3). The resulting *E. coli* expression strains were grown at 37 °C to an optical density at 600 nm of 0.5. At this point, cultures were supplemented with IPTG to a final concentration of 0.5 mM to induce protein expression and then further incubated overnight at 16 °C with shaking (200 rpm). Cells were harvested by centrifugation ($3000 \times g$, 10 min) and resuspended in binding buffer (50 mM NaH2PO4, 300 mM NaCl, pH 8.0). Then, cells were lysed by sonication as above and centrifuged at $16,000 \times g$ for 20 min at 4 °C to remove the membrane fraction and cell debris. Finally, lactylases were purified using Ni-NTA agarose columns as described above.

## Lactylase activity assay

A solution containing 1 μg/ml (final concentration) of the purified recombinant lactylase was incubated with 10 μg/ml (final concentration) of unmodified peptide (VIRTKGTIAG) in reaction buffer (100 mM HEPES, 100 mM MgCl2, 10 mM KCl, 0.1 mM lactyl-CoA, pH = 7) for 1 h at 37 °C[21]. The reaction was stopped with an equal volume of 10% (v/v) trifluoroacetic acid and the enzyme in the sample was removed by ultrafiltration centrifugation using Amicon Ultra-0.5 3 K centrifugal filter devices (Millipore). The lactylated peptide (VIRTK(la)GTIAG) was

used as positive control, and the K-to-R peptide derivative (VIRTRG-TIAG) was used as negative control. For dot blots, 3 µl-samples were dropped on nitrocellulose membranes and membranes were dried at room temperature. See detailed information on blocking and the antibody incubation process in the immunoblotting analysis section.

The peptide reaction products produced by the activity of lactylases 1173 and 2573 were further analyzed by LC-MS/MS. Samples were desalted using C18 ZipTips (Millipore) and dissolved in solvent A (0.1% formic acid in water) and separated with the following gradient to solvent B (0.1% formic acid in acetonitrile) using a NanoElute UHPLC system (Bruker Daltonics) coupled to a timsTOF Pro2 mass spectrometer (Bruker Daltonics): 5–30% in 20 min, 30–45% in 5 min, 45–85% in 2 min, 85% for 3 min, at a constant flow rate of 300 nl/min. One microliter of loading sample was separated on a reversed-phase chromatography column (IonOpticks, Aurora series emitter column C18, 1.9 µm, 250 mm × 75 µm). Survey full-scan mass spectra were acquired across the mass range of 100–1700 m/z in positive electrospray mode with an accumulation ramp time of 100 ms. The single cycle acquisition period was 1.16 s, including one total ion MS scan and ten parallel accumulation serial fragmentation (PASEF) MS/MS scans. To ensure optimal accuracy, a dynamic exclusion period of 30 s was implemented. The mass spectral data were searched online using PEAKS ONLINE (X Build, version 1.4.2020-10-02_113407). The reference database was the sequence of *S. aureus* alpha-toxin (GenBank: CAQ49583.1). "No enzyme" was set as the cleavage mode of peptides. The minimum length of the peptide segment was set to six amino acid residues. The mass tolerance for precursor ions was set to 20 ppm, and the mass tolerance for fragment ions to 0.05 Da. Lactylation of lysine ( + 72.021) was considered as variable modification. A confidence score for peptides and proteins of greater than 20 ($-10lgP \geq 20$) was used.

## Gel electrophoresis and immunoblotting analysis

Cultures were inoculated (1:100) from pre-cultures grown overnight and incubated at 37 °C with shaking until they reached the stationary phase. Subsequently, the bacteria were harvested by centrifugation (3000 × g, 10 min) and adjusted to an optical density of 1 at 600 nm. The cell pellets were then treated with lysostaphin (final concentration, 50 µg/ml) for 30 min at 37 °C. To the supernatants, trichloroacetic acid was added to a final concentration of 20% and supernatants were incubated at 4 °C for 12 h to precipitate proteins. Then, samples were centrifuged (16,000 × g, 4 °C) for 10 min, the supernatant was discarded, and precipitates were washed three times with cold acetone. The concentrated supernatant was then re-dissolved in sample buffer (0.5 M Tris, pH 8.0, 4% SDS). For SDS-PAGE, cell lysate and supernatant samples were mixed with 4× SDS-PAGE loading buffer (TaKaRa), and heated at 100 °C for 10 min.

For dot blots, 3 µl-samples were dropped on nitrocellulose membranes and membranes were dried at room temperature. For immunoblotting of gels, samples were separated on 10% Tris-glycine gels unless otherwise specified, proteins were transferred (25 V, 1.3 A, 7 min) onto polyvinylidene fluoride (PVDF) membranes (0.45 µm, Millipore) using the Trans-Blot Turbo System (Bio-Rad), and blocked using 5% skim milk in Tris-buffered saline with Tween (TBST, 25 mM Tris-HCl (pH 8.0), 150 mM NaCl, 0.1% Tween 20]. The membranes were incubated with antibodies specific for lysine-lactylated proteins (Pan-αKla, PTM Biolabs, PTM-1401RM, RRID: AB_2942013, 1:1000), lysine-acetylated proteins (Pan-αKac, PTM Biolabs, PTM-105RM, RRID: AB_3099509, 1:1000), alpha-toxin (αHla, Abcam, ab190467, RRID: AB_3099508, 1:1000), sortase A (αSrtA, produced by GLbiochem, 1:1000), alpha-toxin lactylated at position K84 [αHlaK84la, produced by PTM Biolabs against peptide VIRTKlaGTIAGRC coupled to keyhole limpet hemocyanin (KLH) and affinity-purified by PTM Biolabs using unmodified peptide to react only with lactylated peptide 1:1000], unmodified Hla (αHla*, produced by PTM Biolabs against unmodified peptide VIRTKGTIAGRC coupled to KLH, 1:1000), ADAM-10 (αADAM-

10, ABclonal, A25167, RRID: AB_3099510, 1:1000), caveolin-1 (αCaveolin, ABclonal, A19006, RRID: AB_2862498, 1:1000)at 4 °C overnight. After incubation with the primary antibodies, the membranes were washed three times in TBST and incubated with the secondary antibodies (anti-rabbit IgG, HRP-linked antibody, Cell Signaling Technology, 7074, RRID: AB_2099233, 1:5000 or anti-mouse IgG, HRP-linked antibody, Cell Signaling Technology, 7076, RRID: AB_330924, 1:5000). Reactions were visualized with an enhanced chemiluminescent (ECL) substrate kit (Thermo Fisher Scientific) and blots were imaged using a Tanon 4200 imaging system. All original blots are shown in the Source Data.

## Coomassie blue staining

Coomassie blue staining was performed using a Biofraw Fast Protein Stain Kit (Tanon). Gels were stained at room temperature for 0.5–1 h. Then, decolorization solution [10% (v/v) acetic acid, 5% (v/v) ethanol] was added and samples were incubated for 2–16 h.

## Cytotoxicity detection

Cytotoxicity was measured by release of lactate dehydrogenase (LDH). Cultures inoculated 1:100 from pre-cultures grown overnight were incubated for 15 h at 37 °C with shaking. A549 cells were incubated for two hours with purified alpha-toxin or bacterial supernatants (1:50 dilution) at 37 °C. Cell lysis was measured using an LDH cytotoxicity assay kit (Roche) following the manufacturer's protocol. The amount of LDH released was normalized to the total LDH content measured in cell samples and Triton-X-100 was added to a final concentration of 1%.

## Pull-down assay

A549 cells ($5 × 10^6$) were lysed using radio immunoprecipitation assay (RIPA) buffer (2 ml). Then the cell lysate was centrifuged at 16,000 × g at 4 °C for 10 minutes. The supernatants were carefully transferred into separate microfuge tubes (200 µl/tube). To one sample of supernatant used as control, loading buffer was added and the sample was boiled for ten minutes. To the other samples, 10 µg of purified alpha-toxin and 1 ml RIPA buffer were added, and samples were incubated in microfuge tubes for 3–24 h at 4 °C with constant tumbling. For the mock group, an equal volume of PBS was added to the cell lysate, and the samples were incubated under the same conditions for 24 h. Then, a 30 µl slurry of Ni-NTA agarose (Qiagen) was added to alpha-toxin and mock sample tubes, and tubes were incubated for 30 min at 4 °C with constant tumbling. The Ni-NTA agarose was spun down by centrifugation at 600 × g at 4 °C for 2 min, washed three times with RIPA buffer, and proteins were eluted with 250 mM imidazole elution buffer.

## Mouse infection models

In all experiments, the health of the mice and disease progression were monitored every day. For the abscess model, BALB/c female mice were anesthetized using 2,2,2-tribromoethanol (dissolved in 2-methyl-2-butanol). Then they received 100 µl of PBS containing $2.5 × 10^7$ (for wild-type, SA-K84R, and Δ1173 + 2573 strains) or $1 × 10^8$ (for Δ*hla* and Δ*hla* + 1173 + 2573 strains) CFU of *S. aureus* (grown to mid-logarithmic phase) or PBS alone by intradermal injection. After 24 hours, the length (L) and width (W) of the abscesses were measured to calculate the area (A) using the formula A = π (L × W)/4.

For the lung infection model, BALB/c female mice (6–8 weeks old, Shanghai JSJ Laboratory Animal Co, Ltd.) were used. Mice were anesthetized with 2,2,2-tribromoethanol, and received 40 µl of PBS containing $2 × 10^9$ (for wildtype, SA-K84R, and Δ1173 + 2573 strains) or $8 × 10^9$ (for Δ*hla* and Δ*hla* + 1173 + 2573 strain) CFU of *S. aureus* (grown to mid-logarithmic growth phase) in group A, and $2 × 10^8$ (for wild-type, SA-K84R, and Δ1173 + 2573 strains) or $8 × 10^8$ (for Δ*hla* and Δ*hla* + 1173 + 2573 strain) CFU in group B, by slow intra-nasal inoculation. All surviving animals in group A were euthanized on the fifth day post inoculation. In group B, mice were euthanized 48 h post

inoculation, and their lungs were dissected. The bronchoalveolar lavage fluid (BALF) was collected for cytokine detection using an ELISA kit (Abcam, IL-1β: ab197742, IL-6: ab100712, TNF-α: ab100747) and leukocyte counting. The left lung was homogenized in 0.5 ml of PBS, and the homogenized tissue was diluted and plated on TSB agar for CFU determination. The right lung was fixed in 4% paraformaldehyde to prepare paraffin sections for H&E and Ly6G staining. The primary antibody used in the immunohistochemical (IHC) staining to detect leukocytes was obtained from Abcam (ab238132, 1:1000). CellSens imaging software (version 4.1) was used to image tissue paraffin sections for H&E and Ly6G staining (Olympus). Slides were examined independently by a histopathologist who was blinded to the treatment. The number and proportion of Ly6G positive cells were analyzed by using Image J (version 1.51j / Java 1.8.0_112 64-bit).

To quantify leukocyte infiltration in the lungs of infected mice, BALF was collected as described above. We used BD Trucount Absolute count tubes (BD Biosciences 340334) to count absolute $CD45^+$ cell numbers in the BALF using PerCP rat anti-mouse CD45 (BD Biosciences, 557235, RRID: AB_396609, 1:100). For leukocyte classification, neutrophils were defined as $CD45^+CD11b^+$ $Ly6G^+$ cells, and macrophages were defined as $CD45^+CD11b^+$ $F4/80^+$ cells. BALF samples were centrifuged ($400 \times g$) for five minutes and cells were resuspended in 100 μl of sterile PBS. Cells were incubated with Fc Block (purified anti-mouse CD16/32 antibody, BioLegend, 101302, RRID: AB_312800, 1:500) and stained with Fixable Viability Stain 450 (BD Biosciences, 562247, RRID: AB_2869405, 1:100), CD45-PerCP (BD Biosciences, 557235, RRID: AB_396609, 1:100), CD11b-FITC (BioLegend, 101206, RRID: AB_312789, 1:100), Ly6G-PE (BD Biosciences, 551461, RRID: AB_394208, 1:100) and F4/80-APC (BioLegend, 123116, RRID: AB_893493, 1:100), and RBCs were removed from the single cell suspension using BD FACS lysing solution (BD Biosciences 349202). Samples were analyzed using NovoExpress software (version 1.5.8) after detection by flow cytometry (NovoCyte Advanteon Dx, Agilent).

## Statistics

Statistical analyses were performed using Graph-Pad Prism for Macintosh, version 9.3.1. Survival curves were compared using the log-rank (Mantel-Cox) test. Unpaired, two-tailed t-tests were used to compare two, and ANOVAs were used to compare differences between variables of more than two groups. Normality of distribution for groups with $n > 3$ was ascertained using Shapiro-Wilk tests. All error bars on the graphs represent the standard deviation (±SD). A $P$ value of less than 0.05 was considered statistically significant. The sample sizes and replicates are described in the corresponding legends. All repeats were obtained from different samples.

## Reporting summary

Further information on research design is available in the Nature Portfolio Reporting Summary linked to this article.

## Data availability

All data needed to evaluate the conclusions in the paper are present in the paper or the supplementary material. The MS proteomics data generated in this study have been deposited in the PRIDE database (Kla proteome data, accession number PXD046022; proteome data, accession number PXD046023, http://www.ebi.ac.uk/pride). Source data are provided with this paper.

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

## Acknowledgements
This work was supported by the National Key Research and Development Program (No. 2022YFC2603800 and 2023YFC2306200 to M.L.), the National Natural Science Foundation of China (82172325 to M.L. and 82102455 to Y.W.), the Shanghai Sailing Program (21YF1425500 to Y.W. and 22YF1438400 to Y.D.), and the Intramural Research Program of the National Institute of Allergy and Infectious Diseases (NIAID, project number ZIA AI000904, to M.O.).

## Author contributions
Conceptualization, M.L. and M.O. Methodology, Y.W. and Yanfeng.L. Investigation, Y.W., Yanfeng.L., G.X., Y.J., Z.Y., T.C., X.M., N.Z., Y.D., Yan.L., H.W., L.H., B.S., Q.L. and Yao.L. Funding acquisition, M.L., Y.W., Y.D., and M.O. Supervision, M.L. and M.O. Writing, M.O.

## Competing interests
The authors declare no competing interests.
