## [Transparent Peer Review file · Nature Communications]

Post-translational toxin modification by lactate controls *Staphylococcus aureus* virulence

Corresponding Author: Dr Michael Otto

This manuscript has been previously reviewed at another journal. This document only contains reviewer comments, rebuttal and decision letters for versions considered at Nature Communications.

Version 0:

Reviewer comments:

Reviewer #1

(Remarks to the Author)

My concerns have been sufficiently addressed in the revisions.

Reviewer #2

(Remarks to the Author)

The authors have partly addressed my concerns however I remain skeptical of their conclusions and the study, although novel, remains in parts preliminary

Reviewer #3

(Remarks to the Author)

The revised version of the study by Wang, et al substantially address the comments from the initial set of reviews, strengthening support for the primary conclusions of the paper. Most importantly, the authors have expanded the in vivo relevance of the findings presented, which are considered to be a substantial contribution to the field.

While the conclusions currently reflect less emphasis on the mechanism by which lactylation of Hla at residue 84 functions, the conclusions drawn from Fig. 4 would still benefit from either additional quantification or modification of the wording in the text. Specifically, the apparent phenotype of the Hla ($\Delta 1173+2573$) compared to the K84R variant as presented in panel c appear nearly identical, both distinct from the wild-type Hla finding. This is distinct from the conclusion drawn when considering the data presented in panel e wherein monomer interaction with A549 cells appears equal across all three variants. In addition, these findings appear to contrast with the observation that the Hla ($\Delta 1173+2573$) exhibits oligomerization at physiologic temperatures that is quite similar to wild-type Hla - very distinct from that observed with the K84R mutant. This latter finding again suggests that the K84R mutant is impaired in biological activity beyond simply lacking lactyl modification. (The results presented in Fig 4c do not aid in addressing this, as the amount of Hla added to the pull-down assay is substantially in excess of that in which physiologic binding interactions between ADAM10 expressed on the cell surface of live cells occur.) It would be beneficial for the authors to revise the conclusions drawn from this facet of the study, especially including those in lines 220-222 which are not congruent with the data presented. Alternatively, it may be better to limit the presentation of this data rather than provide 'preliminary' findings.

Reviewer #4

(Remarks to the Author)

Wang et al. have submitted a revised version of the on lysine lactylation of *S. aureus* alpha-toxin and its influence on virulence of the pathogen. The authors have answered all of my points and provided the minimum information requested (annotated MS/MS spectra of lactylated peptides after in-vitro assay). I agree that failed detection of lactylation on the full-length alpha-toxin by MS may have been caused by low abundance/occupancy of the protein, and suggest to mention this

negative result in the text. From my point of view, the manuscript can be accepted for publication.

REPLY TO REVIEWER COMMENTS

Reviewer #1 (Remarks to the Author):

My concerns have been sufficiently addressed in the revisions.

Reviewer #2 (Remarks to the Author):

The authors have partly addressed my concerns however I remain skeptical of their conclusions and the study, although novel, remains in parts preliminary

Reviewer #3 (Remarks to the Author):

The revised version of the study by Wang, et al substantially address the comments from the initial set of reviews, strengthening support for the primary conclusions of the paper. Most importantly, the authors have expanded the in vivo relevance of the findings presented, which are considered to be a substantial contribution to the field.

While the conclusions currently reflect less emphasis on the mechanism by which lactylation of Hla at residue 84 functions, the conclusions drawn from Fig. 4 would still benefit from either additional quantification or modification of the wording in the text. Specifically, the apparent phenotype of the Hla (Δ 1173+2573) compared to the K84R variant as presented in panel c appear nearly identical, both distinct from the wild-type Hla finding. This is distinct from the conclusion drawn when considering the data presented in panel e wherein monomer interaction with A549 cells appears equal across all three variants. In addition, these findings appear to contrast with the observation that the Hla (Δ 1173+2573) exhibits oligomerization at physiologic temperatures that is quite similar to wild-type Hla - very distinct from that observed with the K84R mutant. This latter finding again suggests that the K84R mutant is impaired in biological activity beyond simply lacking lactyl modification. (The results presented in Fig 4c do not aid in addressing this, as the amount of Hla added to the pulldown assay is substantially in excess of that in which physiologic binding interactions between ADAM10 expressed on the cell surface of live cells occur.) It would be beneficial for the authors to revise the conclusions drawn from this facet of the study, especially including those in lines 220-222 which are not congruent with the data presented. Alternatively, it may be better to limit the presentation of this data rather than provide 'preliminary' findings.

Reply:

We now deleted, as suggested, the mentioned preliminary data and the respective discussion regarding mechanistic aspects of how lactylation influences alpha-toxin at the molecular level (previous Figures 4c and e). We now only left the findings that

indicate it is not receptor-mediated (current Fig. 4c). We hope this addresses the reviewer's concerns and thank the reviewer for recognizing that the focus of our paper was on in-vivo consequences of the mechanism we discovered, while a detailed analysis of how lactylation impacts alpha-toxin oligomerization may be the subject of future investigation.

Reviewer #4 (Remarks to the Author):

Wang et al. have submitted a revised version of the on lysine lactylation of *S. aureus* alpha-toxin and its influence on virulence of the pathogen. The authors have answered all of my points and provided the minimum information requested (annotated MS/MS spectra of lactylated peptides after in-vitro assay). I agree that failed detection of lactylation on the full-length alpha-toxin by MS may have been caused by low abundance/occupancy of the protein, and suggest to mention this negative result in the text. From my point of view, the manuscript can be accepted for publication.

Reply:

After thorough discussion, we decided not to follow that suggestion because we found it unusual to describe failed efforts. As the reviewer indicated that the manuscript can be accepted, we felt that the reviewer did not consider this change mandatory.